# Constraints on the functioning of Community-based Health Planning and Services facilities: A qualitative study in the Jirapa Municipality, Ghana

Roger Kuutero Kaburu[1], Umar Haruna[2], Gilbert Abotisem Abiiro[3,4]*

**1** St. Joseph's Nursing and Midwifery Training College, Jirapa, Upper West Region, Ghana, **2** Department of Social and Behavioral Change, School of Public Health, University for Development Studies, Tamale, Ghana, **3** Department of Health Services, Policy, Planning, Management and Economics, School of Public Health, University for Development Studies, Tamale, Ghana, **4** Department of Population and Reproductive Health, School of Public Health, University for Development Studies, Tamale, Ghana

* gabiiro@uds.edu.gh

## Abstract

The declaration of the Alma-Alta on primary health care (PHC) in 1978 enjoined nations to make health care accessible, affordable, and situated within their cultural contexts. The Ghana Community-based Health Planning and Services (CHPS), as a strategy to achieve the goal of PHC, has shown significant successes in communities where it has been implemented. However, a number of challenges continue to affect the effective functioning of CHPS. This study explored the community level and health system constraints on the effective functioning of CHPS in the delivering of PHC services in the Jirapa Municipality. A qualitative approach was implemented. A criterion-based purposive sampling technique was employed to recruit 51 managers and health service providers of CHPS for key informant interviews. The respondents included 25 community health management committee members, 25 health officers in charge of CHPS facilities, and one municipal CHPS coordinator. The interviews were held from September 18 to November 23, 2020. All interviews were face-to-face, audio-recorded and transcribed verbatim. Thematic analysis based on the constant comparative method was employed to analyse the data. The results showed that low community involvement in CHPS activities, disputes over the location and naming of CHPS zones, inadequate understanding of the CHPS concept and religious beliefs were the key community level factors which negatively affected the functioning of CHPS. Also, lack of logistics, financial constraints, poor attitude of health workers and inadequate staff motivation were the health sector constraints on the effective functioning of CHPS. In conclusion, concerted efforts are needed to tackle the community level and health system constraints to improve the overall functioning and effectiveness of the CHPS strategy. We recommend the strengthening of community sensitization, timely disbursement of funding, and provision of infrastructure and supplies to improve upon the effective functioning of CHPS as a strategy for delivery PHC in Ghana.

**Data Availability Statement:** The study used qualitative data that was tape-recorded and transcribed. The tape recordings and transcripts, which constitute the dataset for this study, cannot

**Funding:** The authors received no specific funding for this work.

**Competing interests:** The authors have declared that no competing interests exist.

## Background

Primary health care (PHC) has been promulgated for over four decades as a global strategy for ensuring basic health care for all people and it is also the cornerstone of the health-related Sustainable Development Goal (SDG) [1]. A substantial chunk of recent health policy discussions is centred on achieving the 2030 Sustainable Development Agenda, a milestone that prioritizes universal health coverage (UHC) [2, 3]. The 9th Global Conference on Health Promotion reaffirmed the role of PHC in achieving the 2030 Sustainable Development Agenda [4, 5]. PHC is *"essential health care based on practical, scientifically sound and socially acceptable methods and technology made universally accessible to individuals and families in the community through their full participation and at a cost that the community and the country can afford"* [6]. PHC supports the goal of 'health for all' by acting as the first point of contact for patients and by providing care that is both family and community oriented, taking into account the critical influences of social networks, and providing services that are well-coordinated and ensure continuity of care [7].

Rural-urban disparities in the availability and accessibility of health care, particularly in poor and middle income nations, have resulted in widespread differentials in morbidity and mortality rates across the world [8]. Rural residents in most of these nations are constrained in their utilization of health services than their urban counterparts due to a variety of socioeconomic and health system constraints [9]. Given the inequalities that occur within countries, reaching UHC would necessitate all-inclusiveness and participatory interactions between health systems and the people [10]. As a result, the performance of community health systems has become increasingly important in both high-income and low-and middle-income countries (LMICs) [4].

The declaration of the Alma-Alta on PHC in 1978 enjoined nations to make health care accessible, affordable, and situated within the cultural context of the people [6]. Inspired by the Alma Ata commitment to PHC, and regarded as a bold departure from bureaucratic models of health service delivery, the Community-based Health Planning and Services (CHPS) is a Ghanaian national health policy initiative that was piloted in 1999 [11, 12]. Under the CHPS programme, community health providers, assisted by community volunteers, are required to deliver healthcare closer to community members' homes, especially in rural and slum communities [11]. Maternal and reproductive health care, child health services, health education, healthy lifestyle promotion, minor ailment case management, community mobilization for health action, referrals, home visits, and client follow-ups are all included in the key service package of CHPS [11, 12]. As a strategy to increase rural access to health care services while empowering local communities to take greater control over their health, the CHPS initiative sought to promote community-driven health care services, with technical support from the Ghana Health Service [13]. CHPS, therefore, aimed to bring health services to community locations, develop sustainable volunteerism and community health action. This invariably empowers women and vulnerable groups, and helps to improve interaction among health providers, members of the households and the community at large [11]. This initiative aims to remove physical and geographical barriers to health care delivery in Ghana's poorest regions and communities [14]. Its ultimate goal is to transform the dynamics of rural health care service delivery from community health care providers who passively wait for patients into outreach workers who actively seek patients in communities and their homes, also known as doorstep services [15]. CHPS implementation at the local level necessitates collaboration between the health sector and communities, as it entails systematic planning and bargaining with all stakeholders, including the local government, political establishment, and community representatives, through community mobilization and effective involvement [14]. The CHPS

programme with its health volunteers, vibrant community involvement, resident nurses, and routine household outreach services in which nurses visited every home a minimum of four times per year, lowered mortality by almost 30% even in communities with relatively low health seeking behaviours [16].

Even though the CHPS strategy showed significant impact in the pilot communities, its replication and scaling up came with a lot of challenges and its impact on health and demographic outcomes has since stagnated [17]. According to the National CHPS Policy, a well-functioning CHPS zone that stands the test of time shall have the following characteristics; a well-functioning supervision system, a well-organized team of health workforce (Community Health Officer [CHO] and Community Health Volunteers [CHVs]), continuous quality health services, a good health financing system, a good leadership and governance system, and a team approach to service delivery [11]. A well-functioning CHPS provides a platform for the provision of integrated PHC to communities. The provision of these services is carried out jointly by the CHO and CHV and supported by the Community Health Management Committee (CHMC) [11]. The CHMC include individuals from communities within the CHPS zone, representing various constituents (youth, women, the aged, etc.) within the catchment area of the CHPS zone. The CHPS service package includes community linkage and outreach services; basic clinical services; and management of activities, logistics, and services.

In spite of its promising start and positive prospects, the performance of the CHPS initiative has since been experiencing continuous deterioration in some aspects of health service delivery; particularly in the aspect of home visits, immunization coverage, antenatal care, child welfare clinics and community participation [16]. This situation is not different in the Jirapa Municipality, where even though there are CHPS facilities dotted everywhere, improving access to quality health care is still considered a major challenge in most communities [18, 19]. In the Jirapa Municipality for instance, childhood vaccination by CHPS for Penta 3, has witnessed a decline from a coverage of 62.3% in 2020 to 62% in 2021 and to 55% in 2022 [20].

Most studies have so far focused on the contribution of CHPS facilities in health care delivery as a benchmark for further expansion of CHPS, community involvement and perceptions of the CHPS strategy for improving health outcomes in Ghana [21, 22] and the lessons learned from scaling up CHPS [15] with limited attention on the factors affecting the effective functioning of CHPS in Ghana. To bridge this gap in literature and generate evidence that is policy relevant to the current issues confronting CHPS, this study examined the community level and health system constraints on the effective functioning of CHPS.

## Methods

### Study setting and approach

The study was conducted in the Jirapa Municipality, located in the North Western part of the Upper West Region of Ghana. The Jirapa District was upgraded to a municipality status on 15th March, 2018 by L.I. 2278. The Municipality has a territorial size of 1,188.6 square kilometres representing 6.4 percent of the regional landmass. The projected total population of the Municipality is 106,670 comprising 52,454 males and 54,216 females distributed across 138 communities. The municipal economy is characterized by agricultural activities, services, agro-processing and other small scale manufacturing activities [23].

The municipality has 57 health facilities comprising seven health centers, 47 CHPS zones, one polyclinic, one private clinic and a hospital. Out of these health facilities, three health centers and one hospital are managed by the Christian Health Association of Ghana (CHAG) [20]. There are also 138 community-based surveillance volunteers (CBVs), 43 herbalist and traditional healers and 11 chemical sellers to facilitate healthcare delivery in the municipality

[20]. The main health challenges of the municipality are, increasing incidence of lifestyle and diet-related diseases, huge gaps in geographical and financial access to quality health care and a limited coverage of social protection interventions [23].

The study used a qualitative approach involving interviews with CHPS stakeholders. The qualitative approach allowed us to obtain rich views from participants in their natural setting. The approach enabled an in-depth exploration of the real-life experiences and perceptions of respondents regarding the effective functioning of CHPS in their communities.

## Study population and sampling

The study population consists of key stakeholders of the CHPS programme within the Jirapa Municipality. The key stakeholders included 25 CHMC members, 25 health officers in charge of CHPS facilities, and one municipal CHPS coordinator. A criterion-based purposive sampling technique was used to select the key stakeholders of CHPS as key informants. CHMC chairpersons, CHPS facility in-charges, and the municipal CHPS coordinator were selected because of their expertise, experience, or position within the society, enabling them to provide rich information and a deeper insight into the issue at hand. The sample of 51 respondents was deemed adequate because the data collection reached a saturation point, hence data collection ended at the point where the responses given by participants started to become repetitive.

## Data collection and analysis

Key informant interviews were used to collect data for this study. An interview guide was developed to capture information related to the factors affecting the effective functioning of CHPS in the area. The interview guide explored two broad themes namely, the community level factors affecting the effective functioning of CHPS, and the health system level factors that hampered the smooth operations of CHPS. A semi-structured instrument consisting of questions and probes to address the objectives of the study was used. Before the commencement of data collection, the interview guide was pre-tested at two CHPS zones in Babile a town which shares similar characteristics with the study area. The instrument was slightly revised for clarity, comprehensiveness and relevance, following the pre-test.

All interviews took place where it was most convenient for participants and lasted between 30–50 minutes each. Interviews were conducted face-to-face at a serene location at the convenience of the participant. A team of three researchers collected the data. The team included the first author and two trained research assistants. The research assistants were graduate students pursuing courses in public health and possessed extensive qualitative research experiences and were familiar with the terrain and the culture of the people of the study area. All data collectors were fluent in the local (Dagaare) language. Some interviews were conducted in English and others in Dagaare. Participants were able to freely speak English or Dagaare, so they could express themselves comfortably. The interviews were audio-recorded, transcribed verbatim and those in Dagaare were translated into English by the researchers. Field notes were documented, reconstructed and expanded following each interview, and the data from the field notes were incorporated into the analyses. Transcripts and notes were kept confidential in password-protected files.

The three authors participated in the data analysis. The transcripts were read independently by each team member to develop a coding structure reflecting issues arising from the data. Coding of the transcripts was done using NVivo, and commonly occurring themes and sub-themes were identified. Thematic analysis relying on the constant comparison approach was adopted. Each author read and re-read each transcript and generated initial themes. The themes generated from the inductive coding of the first transcript served as the basis for

constant comparison of the themes from the coding of the subsequent transcripts. The themes emerging from the coding of the transcripts by the three analysts were discussed jointly by all authors and common themes adopted. Where disagreements occurred, the three analysts reviewed their coding and further discussed the conflicting themes among themselves to reach an agreement. The results of the analysis are presented with supporting quotations from the data to elucidate respondents' voices.

### Ethical considerations

Institutional ethical approval was obtained from the Kwame Nkrumah University of Science and Technology (KNUST) ethics review committee with ethical clearance number CHRPE/AP/453/20 before commencement of the study. An access approval was obtained from the Regional Health Directorate and the Jirapa Municipal Health Directorate as well as the Municipal Assembly before the study was carried out in the Municipality. Again, a written informed consent was obtained from opinion leaders in the study communities and the study participants to guarantee their voluntary participation in the study before the study instruments were administered. Also, all benefits and risks relating to the study were explained to all participants. Permission was duly sought before audio recordings of all sessions. The respondents were also assured of confidentiality of responses. Also, direct quotes from participants are identified by codes (CHPS in-charge, CI; CHMC chairpersons, CP; Municipal CHPS Coordinator, MC). Participation in the study was purely voluntary and no participant was coerced, maltreated or treated without respect. The study was implemented in line with the tenets of the declaration of Helsinki.

## Results

### Summary of themes and subthemes

As illustrated in Table 1, two major factors were identified from the themes generated. These are community level factors and health system factors. On the side of the community level factors, disputes over the location and naming of CHPS zones, low community involvement, misunderstanding of the CHPS concept, and religious and traditional beliefs were the key constraints on the functioning of CHPS in the area. Also, health system constraints on the functioning of CHPS included logistical and financial constraints, poor staff attitudes and motivation.

### Community level factors

**Disputes over the location and naming of CHPS facilities.** The study revealed that some communities disagreed with the location and naming of the CHPS facilities in their zones. The interviews revealed that some communities within certain CHPS zones wanted the CHPS

**Table 1. Summary of themes and subthemes.**

| Main theme | Sub theme |
|---|---|
| Community level factors | • *Disputes over naming and location of CHPS facility*<br>• *Low involvement in CHPS activities*<br>• *Inadequate understanding of CHPS concept*<br>• *Negative traditional and religious beliefs* |
| Health system factors | • *Logistical challenges*<br>• *Financial constraints*<br>• *Poor Staff attitudes*<br>• *Poor staff motivation* |

facilities to be named after their communities and/ located in their communities. Such communities became unhappy when the CHPS facilities were named differently, usually in line with the name of the CHPS zone. These disagreements resulting from the naming and location of the CHPS facilities created disunity among the people within the CHPS zones which negatively affected the smooth operations of the CHPS facilities.

*In this zone, the communities have divergent views regarding the location of the CHPS, whiles others support the current location, some are vehemently opposed to where it is cited. This disagreement has also been extended to the choice of name for the CHPS. The other community wanted their community's name to be used for naming the CHPS but this was ignored and has actually created division among the communities. For example, at Tigbe CHPS, the Tia people wanted their name to come first, and the Gbee people too wanted their name to come first (CHPS In-charges).*

*Actually, in some communities they want the CHPS to have their community's name on the sign post. But the understanding is that, the CHPS name should bear the zone name, and so there have been issues when it comes to naming the CHPS. So sometimes the only thing to do to mitigate the situation is to take away the sign post but the assigned name remained unchanged in our documents (CHPS Coordinator).*

**Low community involvement.** The interviews also revealed that community involvement in CHPS activities was low. This was attributed to the inability of the management of the CHPS facilities to provide refreshment for community members for participating in CHPS activities. The interviews revealed that some community members frequently demanded that the CHOs should buy them pito (a local alcoholic drink) for refreshment before they could participate in CHPS activities.

*Community involvement in this CHPS has become low, and the reason is that anytime they come for meeting they usually complain of our inability to provide them 'water' (pito). I think that has contributed to the low involvement. Because they usually expect that anytime they come for meetings we should buy them pito to refresh them (CHPS In-charges).*

**Inadequate understanding of the CHPS concept.** One important reason for the low participation of community members in CHPS activities aside their request for material rewards from the CHO was the lack of understanding of the CHPS concept by some community members. The key officers of the CHPS initiative including the municipal CHPS coordinator argued that before the establishment of a CHPS facility in an area, the residents of that area are often educated on the CHPS concept including its focus on health promotion and disease prevention. However, the interviews revealed that some community members still often misconstrued the CHPS facility as a health centre, and often demanded curative services and refused to honour referrals from the CHOs to higher level facilities for clinical services that could not be provided by the CHPS facilities. This situation depicts a failure of the authorities to properly drive home the purpose of CHPS to the understanding of the people at the grassroots.

*I think the people have mistaken the CHPS to be a health centre or hospital due to their lack of understanding of the CHPS concept. This is because when they are sick and you (CHO) refer them, they usually do not want to accept the referral, and will give you flimsy excuses hoping that they can be treated at the CHPS zone.*

*(CHPS In-charges)*

*You see, normally before we establish a CHPS zone or construct a CHPS compound in a community, we usually hold series of meetings with the community about the operations of CHPS, its core mandate, what CHPS is all about, but immediately after the CHPS is launched, some members of the community will start to clamour for more curative services than the preventive services. So, they see it as a clinic (health center) and not CHPS and therefore their demands become more of the clinical than the preventive services (CHPS Coordinator).*

*I referred one woman whose child had convulsion; the woman complained that she did not understand why she was being referred to another facility when there was one in her community. She was concerned about how her other children were going to be taken care of if she honoured the referral. She felt I was deliberately refusing to provide her with the needed care within our facility. even though her child was very sick (CHPS In-charges).*

**Religious beliefs and traditional practices.** Another community level factor could be deduced from religious and traditional beliefs and practices, respectively, which create stigmatization for the use of certain services provided under CHPS. It was observed that because of some religious beliefs, certain community members, refused to permit their spouses to utilize certain CHPS services. For instance, because of some reservations about family planning on the basis of religion, some women were apprehensive when it came to utilizing such services of CHPS.

*About half (50%) of the community members are Muslims. Some of them have issues with the concept of family planning. As a result, they do not allow their wives to come to the facility because they believe coming to the facility may expose them to the possibilities of taking up family planning services which go contrary to their beliefs (CHPS In-charges).*

*A woman attended the facility, but when the husband found out, he was outraged and accused the wife of visiting the facility to undertake family planning. When the woman reported to me that the husband was unhappy with her visit, I tried to intervene by sitting together with them to settle the matter amicably. Unfortunately, he was not appeased and still dissuaded the wife from visiting the facility (CHPS In-charges).*

*With family planning, the women do not always want to come because anytime they come especially the adolescent girls, the town boys will be making mockery of them saying in the local dialect "Wo wana ko wa twor pipili" meaning she is coming to take injections to protect herself against pregnancy and because of this name tagging the adolescents girls do not usually want to visit the CHPS compound (CHPS In-charges).*

Additionally, it was discovered that there was a peculiar traditional practice which prohibits children from receiving any immunization until they were five years of age.

*In this community, one particular traditional practice is a challenge. When a woman gives birth, she is not supposed to immunize the child until the child is five years old (CHPS In-charges).*

This practice would often deprive the child of the necessary doses of immunization required to prevent childhood morbidity and mortality.

## Health system factors

**Logistics challenge.** On the logistics challenge, almost all CHPS compounds complained of broken-down motorbikes, lack of fuel, and shortages of antenatal care (ANC) cards and

drugs. The interviews revealed that as a local adaption to the logistic challenge, health workers often use their personal motorbikes to carry out CHPS activities while the Municipal Health Directorate commit to the servicing and fuelling of such motorbikes. The inability of the CHPS compounds to raise money on their own has made it difficult for such facilities to acquire and maintain the needed tools and logistics required for their effective functioning.

> *Hmmm, it is a challenge because all our motorbikes are broken down, and we are using our private motorbikes for the services. The directorate only supports us with fuel and servicing of the motorbikes, and so it is a challenge* (CHPS In-charges).

> *Our major challenge is that we do not have enough motorbikes for the zones. A lot of the CHNs have therefore been compelled to use their own motorbikes to carry out official duties. As a municipal health management team, we are compelled to service the personal motorbikes for them. Unfortunately, fuel have become another thorny issue we have been grappling with. CHPS facilities do not generate enough money to support some of the costs involved in provision of health services* (CHPS Coordinator).

> *We have inadequate supply of both drugs and non-drugs logistics, most especially we have problems with ANC cards. Mostly we have to make photocopies of the cards for the clients and then explain to them the challenge with shortages of the original cards* (CHPS In-charges).

> "......For this our CHPS compound, there is no midwife, no delivery room, we lack essential medicines, we lack emergency transport, there is no light for the compound. All these affect the work of the CHPS" (CHMC, Member).

**Negative staff attitudes.** Another important revelation of this study is that health workers posted to the CHPS facility occasionally exhibited bad behaviours and attitudes. Among other attitudes, community members complained about absenteeism, verbal insults, fraudulent behaviour and general disrespect which amounted to unprofessional conduct and affected health care service delivery. A respondent blamed the poor health worker-community interactions on differences in cultural values and lifestyles between the non-indigene health workers and residents of the local communities.

> *CHPS requires a lot of interaction between health workers and members of the community. Unfortunately, most of our health workers are not indigenes of our communities. Most of them tend to come from urban communities where the lifestyles and values appear very different from what they encounter in these rural parts. This often results in a clash of values. There was a report of a recent incident where the health worker considered the dressing of a community member as shabby and inappropriate and attempted to point that out, but this resulted in some verbal altercation, which nearly became physical* (CHMC, Member)

Other respondents contributing to the point on poor staff attitude alluded to fraudulent conduct and absenteeism of nurses in the following narrations;

> "There was a report of a community health nurse who collected monies from community members under the pretext of registering them for the National Health Insurance Scheme (NHIS) but ended up spending the monies. This unacceptable conduct has sometimes affected community involvement and CHPS performance" (CHPS Coordinator).

> "......Hmmm for me, truly I think the attitude of the nurses in this facility is not the best. This is because in the previous years, we had a particular nurse even though she was the only

*one, she was committed to the work and was always available to work, but now, even though they are many, you can hardly find them in the community doing the work. . ." (CHMC, Member)*

**Financial constraints.** The study revealed that most facilities are in debt due to delayed reimbursement of NHIS claims. This situation is hampering efforts to even make a request from the regional medical stores for essential drugs to keep the CHPS facilities running smoothly.

*For NHIS, whenever we send our claims to them for reimbursement, it takes several months before we even get a response, these delays have affected our service delivery* (CHPS In-charges).

*Shortages of essential medicines for our operations often arise as a result of non-payment of our bills for supplied drugs. The regional medical stores cannot supply facilities in arrears because they also need to settle their debt to the central medical stores. The result is always a shortage of medicines at the CHPS facilities* (CHPS Coordinator)

In addition to delayed disbursement from NHIS, funding for CHPS to enhance their effective functioning was noted as a challenge, various CHPS have indicated lack of funds to support their work because they do not generate any income.

*We do not have our own accounts we operate with. We are two sub-district facilities and share the same account with the mother (supervisory health center) facility, and hence we thus lack drugs and other logistics to attend to patients mostly due to the delay in payment of NHIS claims and so very often patients will come when you say there is no drugs they will be complaining* (CHPS In-charges).

**Poor staff motivation.** The health system factors also included the inability of management of the health system to provide financial and other material motivations to staff of the CHPS facilities. Respondents complained bitterly about the lack of efforts to motivate staff to serve as incentives for better performance. They also indicated that where efforts are made to motivate them, the package tend to be too meagre and represent a mockery of their expectations. The following comments accentuate the feelings of the respondents.

*Hmmmm, for motivation, we don't have any form of motivation from either the community or the municipal health management team* (CHPS In-charge).

*For motivation, at the end of the year, the DHMT (District Health Management Team) usually give us something small token in an envelope like 50.00 to buy stuffs* (CHPS In-charges).

*Motivation from DHMT is bad, it was last years that we got some motivations but you will work and the year will end but common key soap you will not get* (CHPS In-charges).

## Discussion

The CHPS strategy was conceived as a community driven initiative to enhance health service delivery and to bring healthcare to the door steps of the underserved in line with the goals of PHC. After decades of operationalization and series of reviews, it is timely and apt to examine some of the factors challenging the effective functioning of CHPS especially in rural areas.

Low community involvement has been identified as one of the key factors which hamper the effective functioning of CHPS facilities. Community involvement in CHPS, contrary to policy expectations was rather low. This finding is in line with that of Gyang et al as well as Kweku et al who noted poor community participation as a major problem facing health service delivery within CHPS facilities [22, 24]. In practice, CHPS system served as a national mechanism to deliver essential community based health services through the involvement of communities in the planning and delivery of PHC services [21]. However, due to the low community involvement some of the important services provided under CHPS remain underutilized. Agalga et al have identified volunteer attrition, competing economic activities among others as factors contributing to low community participation in the CHPS programme [21]. Apathy among members of the community has also resulted in low coverage of certain activities such as immunization and other services provided by CHPS [22]. As discovered by our study, the inability of the health officers to incentivize community members with a refreshment for participating in CHPS activities has exacerbated community members apathy towards CHPS activities.

The study also showed that inadequate understanding of the CHPS concept by community members has further contributed to low community involvement in CHPS activities. This situation may be as a result of lack of sensitization or communication challenges between the implementers and community members. This finding is in line with a study by Ghana Health Service [11] which showed that lack of communication and engagement of community members on the concept of CHPS have led to community members not understanding the distinction between community-based health services and services at a higher level health facility. Communities therefore expect any health facility to be able to deliver certain clinical care when required, and when those services are not delivered, their utilization of services is gravely affected. The fact that some CHOs were posted to CHPS compounds without adequate training on CHPS implementation seems to worsen this situation [11]. This has culminated in their inability to adequately educate and sensitize the community members to fully understand the operations of the CHPS. This finding has also been alluded to by previous authors [22, 25], that community sensitization activities carried out by CHOs on the concept of CHPS had considerably influenced the utilization of health services by households. The ineffective nature of knowledge transfer between the system and the community as a result of poor sensitisation, fuelling misconceptions about the CHPS concept, has led to some community members refusing to honour referrals when the CHOs at the CHPS recommended them. If stakeholders, therefore, want to see the CHPS facilities function effectively then there is the need to enhance the sensitization of community members to understand and appreciate the concept of CHPS. Effective community level sensitisation can also reshape religious beliefs and traditional practices constraining the functioning of CHPS. This is capable of improving upon the uptake of services delivered by CHPS, especially that of family planning and child hood immunization services, which have been negatively impacted by existing negative religious beliefs and traditional practices.

Another issue impacting the effective functioning of CHPS is reported disagreements regarding the location and naming of CHPS zones. Similar to this, Kweku et al, have also asserted that sometimes the location of CHPS compounds was geographically inconvenient for some community members [22]. This defeats the principles of PHC where clients are supposed to have access to health care services at a timely and affordable manner especially during emergency situations. The greatest brunt of this is felt by pregnant women, since those who cannot access the CHPS facility for services like skilled delivery, may often resort to self-care, or patronage of services of improperly trained health providers, thereby increasing the number of home deliveries and the risk of maternal and infant mortalities [22]. This means that a

timely and permanent resolution of all disputes regarding the location and naming of CHPS zones is required to ameliorate the situation. Health workers proffered that in situations where these disputes emerge, the best remedy to solving them was to site the CHPS facilities in neutral grounds and use generic names for the CHPS zones.

A fundamental health system factor affecting the functioning of CHPS facilities in the Municipality was logistical challenges. Essential items such as; ANC cards, fuel, motorbikes, transportation for referrals and essential drugs like anti-malaria drugs, basic antibiotics, syrups and paracetamol tablets as well as drugs for pregnant women were often reported not available at the CHPS facilities. Our findings agree with that of Nwameme et al who also noted shortage of medicines in CHPS facilities [26]. Equally, Adongo et al found that health service factors such as lack of adequate CHPS infrastructure and logistics have had an effect on the effective and efficient functioning of CHPS compounds [14]. Related to our findings, a recent study also found that, challenges of CHPS included non-availability of basic logistics, diagnostic accessories, refrigerators for storage of drugs and transport logistics (motorbikes and bicycles) [27]. The CHPS policy has prescribed the type of equipment needed in CHPS facilities for effective service delivery [12]. Unfortunately, many of these facilities visited did not have the full complement of the required equipment. The implications are that service delivery and coverage in these zones will tend to decline.

The inadequate logistics at the CHPS zones emanate from poor funding of CHPS [4]. Given that the CHPS initiative was implemented without any clearly defined regular funding source, it has been a challenge for the existing CHPS zones to mobilize enough funding to carry out their activities. The Ghana Health Service has also not enabled the CHPS to generate their own internally generated funds for the purpose of running the facilities. CHPS zones have to rely on their supervisory health centers for funding. A number of studies on the CHPS initiative have, therefore, consistently reported financial constraints as a major challenge to the effective functioning of CHPS [4, 22, 28]. Financial insufficiency of the CHPS facilities has also been attributed to delayed reimbursement of claims by the NHIS. This finding is in line with that of a study conducted by Assan et al [4] which linked funding challenges with CHPS to delayed reimbursement of claims by the NHIS. This delayed in claims reimbursements has been reported to have negative implications for the purchasing power of CHPS and hence the shortages of drugs and other logistics at the CHPS facilities. The delays in disbursement of NHIS funds is caused by multiple factors including but not limited to financial constraints on the part of NHIS, complicated claims processing procedure, fraud and misappropriation of funds [11, 28].

Another key health system constraint reported by our study was poor attitudes of health workers. The study revealed that in some instances, health providers' attitudes towards work were not encouraging and this had adverse effects on service delivery at the CHPS facilities. This finding appears to be in stark contrast with a study in Kentucky, United States of America, by Jaoko, that noted that the majority of health workers showed positive attitudes towards clients [29]. The findings of our study, however, concurs with a study which suggests that the image of health workers, especially nurses, in Ghana and some other Sub-Saharan African countries is not good [30]. According to Wiru et al [8], the attitude of staff to clients could have a telling effect on utilization of health services at the PHC level. The negative attitudes and behaviours reported in this study were in line with claims by Dapaah that, in Ghana, people generally feel nurses are often unfriendly and rude [30]. This is in line with earlier findings where, patients reportedly experienced verbal abuses, rudeness and neglect [31]. Similar to the findings of O'Donnell, this study also found that health facilities opened and closed irregularly due to staff absenteeism [32]. This is contrary to the Ghana Health Service guidelines which requires CHPS facilities to provide 24 hour services throughout the week [12]. The greatest victims of staff absenteeism are mostly pregnant women and children under five, as well as

other vulnerable persons who may require emergency medical care at any time in the CHPS compound. The poor attitudes of staff towards clients can be attributed to cultural clashes between non-indigene health workers and rural residents on one hand and that of poor staff motivation on the other. The non-motivation of staff could be the reason why health workers in these rural settings often absent from work. Therefore, mechanisms to set aside funds to incentivize the health workforce financially can improve their interest in the work, their attitudes towards clients and enhance their performance in delivery services at the CHPS facilities. Buabeng [33] argue that performance does not depend only on knowledge and skills, but on motivation and job satisfaction. Good motivational packages will significantly transform the working environment of a health facility to undertake great strides to improve the effectiveness and quality of the services provided by the health worker.

## Limitations of the study

The study only covered the perspectives of managers and health service providers under the CHPS initiative. The views and perspectives of other stakeholders, especially community level beneficiaries of the CHPS programme are, therefore, missing. Future studies should therefore include the community (service users) perspective on the factors influencing the functioning of CHPS as well.

## Conclusion and recommendations

Community-based health service delivery is undoubtedly an essential component of the health sector of many developing countries. To realize the goals of PHC and the health SDGs, the optimal functioning of CHPS is vital. The most important community level factors that constrain the effectiveness of CHPS in delivery PHC services include low community involvement in CHPS activities, inadequate knowledge on the CHIPS concept, religious beliefs, and dispute on the location and naming of CHPS compounds. Associated with these are the health system constraints which include lack of logistics, poor attitude of health workers, inadequate funds as well as lack of staff motivation. We recommend a holistic approach to tackling these constraints to ensure that CHPS functions effectively and efficiently. First, at the community level, community sensitization and engagement should be deepened to encourage participation and support for CHPS activities. Thus, health managers and planners at the district level should continuously engage various stakeholders at the grassroots level regarding CHPS' implementation and its milestones. This will enhance community buy in, build capacity of stakeholders and beneficiaries, improve knowledge, and whip up the necessary support for CHPS and its activities. The continuous engagement of various stakeholders will also reduce the apathy toward CHPS, encourage participation, reduce disputes, and improve the knowledge and understanding of the people on the goal and objectives of CHPS. Secondly, policy makers especially the Government should ensure that funding for CHPS is appropriately streamlined and disbursed timely. All sources of funding for CHPS including that of the NHIS should be harmonized and managed efficiently. Provision of logistics, infrastructure and supplies meant for CHPS should be prioritized. District health management teams together with the regional medical stores' managers should formulate and enact actions that will ensure the regular stocking of CHPS zones with essential medicines to help meet the needs of community members and build trust in CHPS activities.

## Acknowledgments

The authors acknowledge the support of Mr Lius Mornaa of Lawra Municipal Health Directorate Mr Benedict Benloo Doozie and Mr. Emmanuel Ghansah-Quarshie of St. Joseph's Midwifery Training College during the data collection.

## Author Contributions

**Conceptualization:** Roger Kuutero Kaburu, Gilbert Abotisem Abiiro.

**Data curation:** Roger Kuutero Kaburu.

**Formal analysis:** Roger Kuutero Kaburu, Gilbert Abotisem Abiiro.

**Investigation:** Roger Kuutero Kaburu.

**Methodology:** Roger Kuutero Kaburu, Umar Haruna, Gilbert Abotisem Abiiro.

**Supervision:** Gilbert Abotisem Abiiro.

**Validation:** Umar Haruna, Gilbert Abotisem Abiiro.

**Writing – original draft:** Roger Kuutero Kaburu.

**Writing – review & editing:** Umar Haruna, Gilbert Abotisem Abiiro.

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
