## [Decision Letter · Decision Letter 0]

17 Apr 2023

PGPH-D-23-00302

Constraints on the functioning of Community-based Health Planning and Services facilities: A qualitative study in the Jirapa Municipality, Ghana

Dear Gilbert Abotisem Abiiro,

Thank you for submitting your manuscript to PLOS Global Public Health. After careful consideration, we feel that it has merit but does not fully meet PLOS Global Public Health’s publication criteria as it currently stands. Therefore, we invite you to submit a revised version of the manuscript that addresses the points raised during the review process.

We look forward to receiving your revised manuscript.

Kind regards,

Hemant

--

Hemant Deepak Shewade, MBBS MD PhD

Academic Editor

Journal Requirements:

Additional Editor Comments (if provided):

Please revise your manuscript keeping in mind comments by both the reviewers, especially Reviewer 1. Do note that both reviewers have not only provided comments, but also attached the submitted pdf along with specific detailed comments using sticky notes option. Authors are requested to note these comments and address them in the point by point response. 

Reviewers' comments:

Reviewer's Responses to Questions

**Comments to the Author**

1. Does this manuscript meet PLOS Global Public Health’s publication criteria? Is the manuscript technically sound, and do the data support the conclusions? The manuscript must describe methodologically and ethically rigorous research with conclusions that are appropriately drawn based on the data presented.

Reviewer #1: Partly

Reviewer #2: Yes

2. Has the statistical analysis been performed appropriately and rigorously?

Reviewer #1: N/A

Reviewer #2: N/A

3. Have the authors made all data underlying the findings in their manuscript fully available (please refer to the Data Availability Statement at the start of the manuscript PDF file)?

Reviewer #1: No

Reviewer #2: No

4. Is the manuscript presented in an intelligible fashion and written in standard English?

Reviewer #1: No

Reviewer #2: No

5. Review Comments to the Author

Reviewer #1: 1. Please provide further details about the Jirapa municipality, specifically about the utilization and status of scale up of CHPS - the rationale should justify why this enquiry now and why here?

2. What is the composition of CHMC? Do they include community stakeholders? How diverse is it? Is there representation of women and members of all ethnic sects? Further details needed

3. "This sampling method was used since qualitative researchers often depend on small sample sizes since

data saturation was the ultimate goal in this type of research, not the number of participants" - This

sampling method was used since qualitative researchers often depend on small sample sizes since

data saturation was the ultimate goal in this type of research, not the number of participants.

4. Conditions/criteria for sample selection has been mentioned partially - would you still call it purposive sampling or would you like to define the sampling method using a more refined terminology? Such as criterion sampling or even theoretical sampling? Refer Qualitative Inquiry and Research Design: Choosing Among Five Approaches

Book by Cheryl N. Poth and John W. Creswell for further information.

5. Who did the data collection?

Please give a brief description of the background and positionality of the researchers who did the data collection and/or the analysis. The constructivist paradigm mandates that the researchers reflexively engage with the research process and hence, a brief description of one's background, orientation and positionality is needed.

6. Please provide some more details about the analysis process. for instance,

1. Did you use constant comparison to move from open  axial  selective codes.

2. How many people were involved in the coding process?

3. Were the memos discussed within the teams?

4. How were the discordances sorted?

5. Since you committed to grounded theory approach, did you use theoretical sampling also? If yes, what were the initial theoretical categories?

6. You have also committed to a constructivist epistemology, indicating that you are closer to the modified approaches of grounded theory rather than the classical approach. In that case, the researchers' theoretical orientation in co-constructing the knowledge is of significance. Please elaborate on this. Refer the following paper for further information:

https://journals.sagepub.com/doi/full/10.1177/2333393618799571#:~:text=There%20are%20three%20primary%20approaches,choices%20when%20using%20grounded%20theory

7. The reasons for low community involvement could have been probed further. This would have required a detailed probing of the community level stakeholders and also the system level stakeholders who are experienced richly in the context of Jirapa municipality.

8. Why did this "lack of understanding of CHPS occur in the community". This would have been an extremely useful enquiry. How did the community develop the "inadequate understanding"? Who transferred the information about CHPS to the community? Were they trained? What modalities did they used for information transfer? Was the process of information transfer adequate in content coverage and appropriate in terms of the mode of delivery? If such critical questions are not asked, the findings would come across as "victim blaming"

9. The failure of authorities to properly drive home the purpose of CHPS to the community could have been highlighted with evidence if the community's side of the story could also have been presented in a nuanced manner. Isn't that the ultimate purpose of qualitative research? To paint a holistic and balanced picture of any phenomenon?

10. The inadequate acknowledgement of religious beliefs and traditional practices in the design of CHPS is an important piece of contextual information which should have informed the implementation of CHPS in this community. Did the system not anticipate this aspect? If not why?

11. Why do these logistic challenges occur?

How does the system cope?

Have they tried any remedial measures?

Have these efforts, if any, produced any results, even if they are localized and of low intensity?

12. Negative staff attitudes is an important perception that can explain the low utilization by the community. However, in order to build a balanced construction, we need to know what do the system stakeholders think about this perception. How do they make sense of this perception. If they agree that this perception by the community is valid or justifiable, then why do they think this situation exists. A balanced epistemological construction requires us to take the knowledge generated from the one category of stakeholders (system) to another category (community) and vice versa for probing.

13. Why does delayed disbursement of NHS claims happen? Is it because of financial constraints at the NHIS or logistic/procedural reasons? The authors can attempt to build potential reasons for this, either from direct interviews with the NHIS level stakeholders or from policy documents or from existing literature.

14. What kind of motivation do the staff actually expect? The descriptions are too trimmed, sanitized and simplistic. Motivation of human resources is an intrinsically complex phenomenon and the authors seem to shy away from delving into the "mess"

15. Discussion is one of the better written sections of the paper.However, the section can be trimmed and structured better to bring out the "so what" of the findings. Also, it is important that the discussion section brings together all the key findings of the paper and build a coherent narrative/theory for the relatively low utilization of the programme.

I have presented a narrative below to consider:

1. Low community involvement has been identified as one of the key factors which hamper the utlization of the scheme.

2. The relatively low understanding of the scheme by the community and the gaps in the programme design characterized by its poor alignment with the social context, stratification, beliefs and value systems of the community could be the potential reasons for this.

3. The negative attitudes of the system level HR perceived by the community indicates the ineffective nature of knowledge transfer between the system and the community, which could in turn be driven by the financial constraints faced by the system and the poor motivation of the HR.

What are the potential reasons for poor community involvement/participation of any community based health programme? What does international literature say about this? Connect this to each step of the aforementioned narrative.

16. In the discussion section, it is critical to connect the negative staff attitudes to their poor motivation and acknowledge the systemic causes for this. Otherwise it looks like we are unjustly labelling the health workers working in a public sector which is marred by financial constraints. The construction should be made more balanced to avoid "blaming"

17. Findings cannot be generalized is NOT a limitation of qualitative research. Qualitative research is built on constructive/transformational paradigms which are based on the ontological and epistemological assumptions that reality is multiple and is socially constructed. Here, our objective is to build balanced and holistic constructions of "how" and "why" social phenomena occur. Please revisit the statement.

18. Overall copyediting and English correction is recommended

Reviewer #2: 1) The manuscript requires copyediting. Sentence formation need to be changed in certain places which are marked with comments in the uploaded reviewer document.

2) In methodology portion few things need to be elaborately described particular places are marked in the file.

3) Stating generalizability as a limitation is not acceptable as qualitative study is not meant for generalizability.

4) All the comments in the file may be addressed for the fine tuning of the manuscript.

6. PLOS authors have the option to publish the peer review history of their article (what does this mean?). If published, this will include your full peer review and any attached files.

**Do you want your identity to be public for this peer review?** For information about this choice, including consent withdrawal, please see our Privacy Policy.

Reviewer #1: No

Reviewer #2: No

---

## [Decision Letter · Decision Letter 1]

11 Jul 2023

PGPH-D-23-00302R1

Constraints on the functioning of Community-based Health Planning and Services facilities: A qualitative study in the Jirapa Municipality, Ghana

Dear Dr. Abotisem Abiiro,

Thank you for submitting your manuscript to PLOS Global Public Health. After careful consideration, we feel that it has merit but does not fully meet PLOS Global Public Health’s publication criteria as it currently stands. Therefore, we invite you to submit a revised version of the manuscript that addresses the points raised during the review process.

Please submit your revised manuscript and the point-by-point response to the reviewers by 25 July 2023. If you will need more time than this to complete your revisions, please reply to this message or contact the journal office at globalpubhealth@plos.org. Please include the following items when submitting your revised manuscript:

We look forward to receiving your revised manuscript.

Kind regards,

Jeannine Uwimana-Nicol, Ph.D.

Academic Editor

Journal Requirements:

Additional Editor Comments (if provided):

Reviewers' comments:

Reviewer's Responses to Questions

**Comments to the Author**

1. If the authors have adequately addressed your comments raised in a previous round of review and you feel that this manuscript is now acceptable for publication, you may indicate that here to bypass the “Comments to the Author” section, enter your conflict of interest statement in the “Confidential to Editor” section, and submit your "Accept" recommendation.

Reviewer #1: All comments have been addressed

Reviewer #2: All comments have been addressed

2. Does this manuscript meet PLOS Global Public Health’s publication criteria? Is the manuscript technically sound, and do the data support the conclusions? The manuscript must describe methodologically and ethically rigorous research with conclusions that are appropriately drawn based on the data presented.

Reviewer #1: Yes

Reviewer #2: Yes

3. Has the statistical analysis been performed appropriately and rigorously?

Reviewer #1: N/A

Reviewer #2: N/A

4. Have the authors made all data underlying the findings in their manuscript fully available (please refer to the Data Availability Statement at the start of the manuscript PDF file)?

Reviewer #1: No

Reviewer #2: Yes

5. Is the manuscript presented in an intelligible fashion and written in standard English?

Reviewer #1: Yes

Reviewer #2: Yes

6. Review Comments to the Author

Reviewer #1: • There are no unjustified claims about the methodology employed

• Sufficient details have been provided about the methods used

• Conclusions are balanced and justified

• Benefitted from an overall copyediting

• Statements which undermined the paradigm have been removed

• I recommend the paper for publication

Reviewer #2: (No Response)

7. PLOS authors have the option to publish the peer review history of their article (what does this mean?). If published, this will include your full peer review and any attached files.

**Do you want your identity to be public for this peer review?** For information about this choice, including consent withdrawal, please see our Privacy Policy.

Reviewer #1: No

Reviewer #2: No

---

## [Decision Letter · Decision Letter 2]

28 Nov 2023

Constraints on the functioning of Community-based Health Planning and Services facilities: A qualitative study in the Jirapa Municipality, Ghana

PGPH-D-23-00302R2

Dear Abiiro,

We are pleased to inform you that your manuscript 'Constraints on the functioning of Community-based Health Planning and Services facilities: A qualitative study in the Jirapa Municipality, Ghana' has been provisionally accepted for publication in PLOS Global Public Health.

Best regards,

Collins Otieno Asweto, PhD

Academic Editor

Reviewer Comments: Take note of Reviewer 3 comments

Reviewer's Responses to Questions

**Comments to the Author**

1. If the authors have adequately addressed your comments raised in a previous round of review and you feel that this manuscript is now acceptable for publication, you may indicate that here to bypass the “Comments to the Author” section, enter your conflict of interest statement in the “Confidential to Editor” section, and submit your "Accept" recommendation.

Reviewer #3: All comments have been addressed

Reviewer #4: All comments have been addressed

2. Does this manuscript meet PLOS Global Public Health’s publication criteria? Is the manuscript technically sound, and do the data support the conclusions? The manuscript must describe methodologically and ethically rigorous research with conclusions that are appropriately drawn based on the data presented.

Reviewer #3: Yes

Reviewer #4: Partly

3. Has the statistical analysis been performed appropriately and rigorously?

Reviewer #3: N/A

Reviewer #4: Yes

4. Have the authors made all data underlying the findings in their manuscript fully available (please refer to the Data Availability Statement at the start of the manuscript PDF file)?

Reviewer #3: Yes

Reviewer #4: Yes

5. Is the manuscript presented in an intelligible fashion and written in standard English?

Reviewer #3: Yes

Reviewer #4: Yes

6. Review Comments to the Author

Reviewer #3: Dear Editor,

Thanks for the opportunity to review this manuscript. My recommendations are below.

1) The title should meet the SMART requirements.

2) Researchers should write down the inclusion and exclusion criteria in more detail.

3) The result part is just putting out the idea; what about the idea saturation issue? You have to express

4) In the recommendation part, it is general; it should be specific to the concerned body.

5) References should be up-to-date, old references, for example, reference number 6,

Best regards

Reviewer #4: Consider reducing the number of quotes from the participants and also avoiding generalizations in some of the quotes, for example, quoting facility coordinators as if they mentioned something yet it was mentioned by few people.

Justification for the setting need to come out clearly, for example, why did you chose that setting?

Justification for the study methodology should be made explicitly.

7. PLOS authors have the option to publish the peer review history of their article (what does this mean?). If published, this will include your full peer review and any attached files.

**Do you want your identity to be public for this peer review?** For information about this choice, including consent withdrawal, please see our Privacy Policy.

Reviewer #3: No

Reviewer #4: **Yes: **Dr Andrew Likaka
